# B7-H3 in Pediatric Tumors: Far beyond Neuroblastoma

**DOI:** 10.3390/cancers15133279

**Published:** 2023-06-21

**Authors:** Cristina Bottino, Chiara Vitale, Alessandra Dondero, Roberta Castriconi

**Affiliations:** 1Department of Experimental Medicine (DIMES), University of Genoa, 16132 Genoa, Italy; chiara.vitale@edu.unige.it (C.V.); alessandra.dondero@unige.it (A.D.); roberta.castriconi@unige.it (R.C.); 2Laboratory of Clinical and Experimental Immunology, IRCCS Istituto Giannina Gaslini, 16147 Genova, Italy

**Keywords:** pediatric tumors, B7-H3, immunotherapy, clinical trials

## Abstract

**Simple Summary:**

Children affected by high-risk tumors receive aggressive standard therapies that significantly improve their survival. Despite this, they have a low quality of life, suffer from life-threatening side effects, and a still-relevant number of them show resistance to therapy and develop fatal relapses. Immunotherapy showed exciting results and is now a promising treatment for cancer patients; targeting B7-H3 fits this scenario. B7-H3 is expressed by many cancers, promotes their growth, and blocks the antitumor responses mediated by cells of the immune system. Promising preclinical results were obtained using antibodies or genetically engineered T lymphocytes recognizing B7-H3 and, currently, different clinical trials are ongoing. Hopefully, the targeting of B7-H3 will help cure adult and pediatric cancer patients.

**Abstract:**

B7-H3 is a 4Ig transmembrane protein that emerged as a tumor-associated antigen in neuroblastoma. It belongs to the B7 family, shows an immunoregulatory role toward NK and T cells, and, therefore, has been included in the growing family of immune checkpoints. Besides neuroblastoma, B7-H3 is expressed by many pediatric cancers including tumors of the central nervous system, sarcomas, and acute myeloid leukemia. In children, particularly those affected by solid tumors, the therapeutic protocols are aggressive and cause important life-threatening side effects. Moreover, despite the improved survival observed in the last decade, a relevant number of patients show therapy resistance and fatal relapses. Immunotherapy represents a new frontier in the cure of cancer patients and the targeting of tumor antigens or immune checkpoints blockade showed exciting results in adults. In this encouraging scenario, researchers and clinicians are exploring the possibility to use immunotherapeutics targeting B7-H3; these include mAbs and chimeric antigen receptor T-cells (CAR-T). These tools are rapidly evolving to improve the efficacy and decrease the unwanted side effects; drug-conjugated mAbs, bi–tri-specific mAbs or CAR-T, and, very recently, NK cell engagers (NKCE), tetra-specific molecules engaging a tumor-associated antigen and NK cells, have been generated. Preclinical data are promising, and clinical trials are ongoing. Hopefully, the B7-H3 targeting will provide important benefits to cancer patients.

## 1. Introduction

B7-H3 (CD276) is a surface glycoprotein belonging to the B7 family. It emerged for the first time as a molecule expressed by tumors in a study published in 2004 [1]. A monoclonal antibody (mAb) (termed M5B14) was generated immunizing mice with a human neuroblastoma (NB) cell line; it had a bright surface reactivity with neuroblasts infiltrating the bone marrow of stage IV (now stage M) patients and stained cancer cell lines of different histotype including glioblastoma and myeloid leukemia. Thanks to a biochemical and proteomic approach, the molecule recognized by the M5B14 mAb was purified and sequenced; the analysis showed its identity with the B7-H3 transmembrane molecule with four Immunoglobulin-like domains (4Ig-B7-H3) described by P. Steinberg and colleagues on myeloid cells [2]. B7-H3 also showed an alternative spliced 2Ig-B7-H3 isoform, which is the only B7-H3 form expressed in mice [3]. The 4Ig-B7-H3 is the form predominantly expressed in humans (hereafter termed B7-H3), is encoded by a gene located on chromosome 15, and results from exon duplication [4]. The study published in 2004 also demonstrated the immunomodulatory role of B7-H3 in humans, in line with what was described in mice [5,6]. B7-H3 inhibited the cytolytic activity of human NK cells against B7-H3 cell transfectants and neuroblasts isolated from bone marrow aspirates of stage M NB patients. Neuroblasts from patients were characterized by high expression of B7-H3, low/null levels of HLA class I, and reduced expression of adhesion molecules and activating ligands; the latter included the Poliovirus Receptor (PVR, CD155) that was crucial for the NK-mediated killing of patients’ derived NB cells [7]. Additional studies confirmed the ability of B7-H3 to inhibit the function of NK and T cells [8] through the engagement of a still unknown inhibitory receptor; thus, B7-H3 has been included in the list of immune-checkpoint ligands [9,10]. We cannot exclude, however, that, as with other members of the B7 family, a stimulatory B7-H3 counterpart may exist [11]. In this context, in mice but not in humans, the 2IgB7-H3 engaged the activating receptor TREM-like transcript 2 (TREML2, TLT-2), expressed by activated T cells and myeloid cells [12].

B7-H3 function in tumors goes beyond immune modulation [13]. In NB and other types of cancer, B7-H3 showed pro-tumoral functions independent of its immune regulatory role; it activates different intracellular pathways, including STAT3 and PI3K, and promotes tumor proliferation, invasiveness, migration, glycolysis, and drug resistance. All these properties account for the negative prognostic role of B7-H3 in many types of cancer including NB [10,14]. For example, the high B7-H3 surface expression in primary tumors from NB patients, in terms of both intensity and percentage of positive cells, correlated with a lower probability of event-free survival. This was true also for patients with localized diseases, thus identifying a group of patients requiring a more careful follow-up [15]. It is of note that soluble B7-H3 forms (sB7-H3) exist that are generated by proteolytic shedding or alternative splicing; sB7-H3 has an important role as a biomarker since high levels are found in cancer patients and correlate with advanced stage of the disease [16].

B7-H3 represents a suitable target for immunotherapy being expressed by tumors of various histotypes [17,18]. Importantly, it is also expressed in the tumor microenvironment and, in particular, by tumor-associated vasculature [19], tumor-associated macrophages (TAM) [20], and cancer-associated fibroblast (CAF) [21]; on the contrary, it is absent or expressed at very low levels on the cell surface of most healthy tissues including normal nervous tissues, and vessels. The B7-H3 expression in vessels and cells populating the tumor microenvironment further makes this molecule a promising therapeutic target. For example, TAM and CAF are present in the NB microenvironment and show pro-tumoral functions related to PGE2 and IL-6 release [22]. Targeting of B7-H3 could also overcome resistance to the standard anti-GD2 immunotherapy in NB children with GD2-negative or GD2-low tumor variants, who represent a relevant percentage (about 12%) of newly diagnosed patients [23,24]. Drug-conjugated anti-B7-H3 mAbs [25] and chimeric antigen receptor (CAR)-transduced T cells [26] targeting B7-H3 have been explored in preclinical studies with very promising results and different clinical trials are ongoing in NB patients [27].

In this encouraging scenario, this review aims at discussing B7-H3 as a possible therapeutic target in pediatric tumors, solid or hemopoietic, other than the prototypical NB, providing an updated and comprehensive overview of preclinical results and ongoing clinical trials (Table 1).

## 2. Tumors of the Central Nervous System

Tumors of the central nervous system (CNS) are the second most common malignancies in childhood. Among these, high-grade gliomas and medulloblastomas are characterized by high morbidity and mortality due to the local invasion and ability to metastasize outside the CNS. The current clinical treatment is aggressive and combines neurosurgical removal of the tumor, craniospinal irradiation, and chemotherapy. Besides the modest quality of life and long-term side effects, the late diagnosis, metastasis, and high rate of relapses result in poor survival chances for patients.

Gliomas are tumors thought to arise from neuroglial stem or progenitor cells that can be localized in cerebral hemispheres, thalamus, brainstem, and spinal cord. In 2021 WHO differentiated pediatric gliomas from adults and the term “glioblastoma” has been abandoned in pediatric oncology [28]. Pediatric high-grade gliomas (pHGGs) are grade III and IV aggressive tumors that include four subgroups: diffuse midline glioma, H3 K27-altered; diffuse hemispheric glioma, H3 G34-mutant; diffuse pediatric-type high-grade glioma, H3-wildtype and IDH wildtype; and infant-type hemispheric glioma. Medulloblastomas (MB) are embryonal tumors of the cerebellum. Four subgroups have been defined depending on the molecular features and prognosis: wingless (WNT, about 10% of cases, favorable prognosis); sonic hedgehog (SHH) and group 4 (about 30% and 35%, respectively, intermediate outcomes); group 3 (about 25% of cases, poor prognosis) [29].

During recent years, different molecules have been discovered to be expressed by CNS tumors that represent promising targets of novel therapies, in particular immunotherapies. Targeting of tumor-associated antigens (TAA) through monoclonal antibodies (mAbs) showed limited results in brain tumors due to the presence and complexity of the blood-brain barrier (BBB). Failures include the mAb-mediated blocking of the immune-checkpoint receptor PD-1 or its ligand PD-L1; these approaches showed remarkable results in adult solid tissues such as melanoma [30], non-small cell lung cancers, and colon and breast cancers. This opened the question of how to treat CNS tumor patients with innovative immunotherapies. A possible strategy is the use of T lymphocytes genetically engineered with chimeric receptors (CAR-T) that seem to be able to cross the BBB, trafficking from blood to brain tissues and cerebrospinal fluid [31]. CAR-T can be systemically delivered or locally administered [32]; the latter approach would increase the effectiveness of treatments, bypassing the BBB and decreasing systemic side effects such as cytokine release syndrome (CRS) or immune effector cell-associated neurotoxicity syndrome (ICANS).

Both gliomas and medulloblastomas express B7-H3 [15,33,34]. In adult glioblastoma, B7-H3 was shown to be highly expressed also in putative cancer stem cells, and B7-H3-CAR-T cells were able to control tumor growth in xenograft murine models [35]. Therefore, B7-H3 represents a promising TAA and different clinical trials targeting B7-H3 are ongoing in adult patients affected by glioblastomas or CNS metastasis, and, more recently, in pediatric patients with recurrent or refractory gliomas and medulloblastomas. Some studies utilize the Iodine-131 or lutetium-177 omburtamab, a radiolabeled B7-H3-specific murine mAb (8H9) that has utility in radio imaging and radio immunotherapeutic uses. They have been used for the treatment of children affected by glioma, medulloblastoma, or CNS-metastasizing neuroblastoma (Table 1). Although some study demonstrates tolerable toxicity and some therapeutic effect, the use of a murine mAb raises doubts, and in 2022 the FDA’s Oncologic Drugs Advisory Committee stopped the use of I131-omburtamab in children with CNS or leptomeningeal metastases from neuroblastoma.

Two promising Phase 1 studies from the Seattle Children’s Hospital are in progress. The first study is recruiting children or young adults affected by CNS tumors including glioma and medulloblastoma. Patients will be treated with autologous T cells expressing B7-H3-specific CAR and a truncated form of the EGF receptor (EGFRt) (ClinicalTrials.gov Identifier: NCT04185038). EGFRt is non-immunogenic, lacks the EGF-binding domain and signaling tail, but retains the extracellular epitope recognized by an anti-EGFR antibody; it is useful as a tracking and selection marker and as a safety/kill switch. B7-H3.EGFRt-CAR-T cells will be delivered via an indwelling catheter into the tumor resection cavity or ventricular system. The second study will recruit children and young adults with diffuse intrinsic pontine glioma, diffuse midline glioma, and recurrent or refractory CNS tumors. The therapeutic is represented by autologous T cells transduced to express combinations of B7-H3, HER2, and EGFR806 CARs, and IL-13-zetakine, which will be locally delivered (ClinicalTrials.gov Identifier: NCT05768880). EGFR806 is a tumor-restricted EGF receptor epitope [36] and IL-13-zetakine is a mutated cytokine (E13Y) that selectively binds to IL-13Ralpha2, a glioma-restricted cell-surface epitope not detectable in the CNS [37]. The use of genetically engineered lymphocytes carrying CAR combinations is appealing; it can improve the therapeutic efficacy of CAR cells and overcome the tumor escape mechanisms downregulating the molecular target selected for immunotherapy. One successful example is represented by CD19/CD20 bi-specific CAR-T cells that prevented antigen escape by malignant B-cell lymphomas [38]. Moreover, very recently, B7-H3-CAR-T cells co-expressing CCR2b, the CCL2 receptor, have been generated that have enhanced the capability of passing the BBB and attacking brain tumor lesions [39].

## 3. Sarcomas

Osteosarcoma (OS), although rare, is the most common bone tumor with a peak incidence in adolescents (18 years) [40]. The knee and humerus are the most common primary sites. It is believed to arise from a multipotent mesenchymal precursor, which generates tumors with different dominant matrices, chondrogenic, fibroblastic, osteoblastic, and telangiectatic [41]. Although localized tumors are treated with surgical therapy alone, those classified as high-grade require combined aggressive treatments because of their ability to metastasize in other bones, lymph nodes, and lungs. Interestingly, extracellular vesicles released by osteosarcoma cells are thought to manipulate the metastatic environment preparing the niche to host migrating tumor cells [39]. Different cancer susceptibility genes have been identified including *RB1* and *TP53* tumor suppressor genes. The *TP53* loss of function is detectable in >90% of osteosarcomas and patients with the Li-Fraumeni syndrome have a high risk of developing osteosarcoma [42]. Accordingly, the introduction of *TP53* mutations into partially differentiated osteogenic stem cells generated osteosarcoma-like cells in vitro [40]. As in other solid tumors, the clinical responses to immune-checkpoint inhibitors (ICIs) were generally disappointing. On the other hand, different targetable surface molecules have been identified including HER2, GD2, and B7-H3.

The Ewing sarcoma (EwS) is a malignant tumor mainly diagnosed in adolescents and young adults [43]. The primary tumor occurs in bones (pelvis, femur, tibia, and ribs) or soft tissues (thoracic wall, gluteal muscle, pleural cavities, and cervical muscles). EwS possesses one of the lowest mutation rates among cancers [44]. The main drivers are pathognomonic chromosomal translocations resulting in the fusion of members of the FET and ETS families; the former are RNA-binding proteins encoded by *FUS*, *EWSR1*, and *TAF15* genes and involved in transcription and splicing; the latter are transcription factors involved in cell proliferation, cell differentiation, cell-cycle control, angiogenesis, and apoptosis. *EWSR1-FLI1* is the most common translocation (>80% of cases). These translocations were detected decades ago, and there is an increasing understanding of their oncogenic role in EwS. Despite this, therapy for patients with aggressive metastasizing tumors still consists of conventional modalities, including, surgery, radio, and chemotherapy.

Rhabdomyosarcoma (RMS) is the most common soft tissue sarcoma in children and young adults <21 years [45]. The primary tumor can arise in a wide variety of anatomical sites and have two typical features, the “alveolar” (ARMS) characterized by cells distributed around an open central space, and the “embryonic” (ERMS) with cells resembling immature skeletal myoblasts. ARMS typically arises in the extremities while ERMS most commonly arises in the head and neck, or genitourinary sites. Although the incidence of ARMS remains constant throughout childhood and adolescence, that of ERMS shows two peaks, in early childhood and early adolescence. Studies have begun to identify factors that contribute to this malignancy [45]. The best characterized are the translocations occurring in *ARMS*, which result in fusion proteins composed of paired box proteins (PAX 3 or 7) and forkhead box protein O1 (FOXO1) [46]. The fusion proteins dysregulate multiple cellular pathways and the *PAX3-FOXO1* gene transfer into mammalian cells was associated with transforming activity, anchorage independence, and loss of proliferation contact inhibition [47]. The cell of origin is not well characterized; different studies suggest that, depending on the anatomical site, RMS could originate from different cell types programmed during tumor transformation toward skeletal myocyte-like cells. Additionally, there is little consensus regarding the risk stratification that, for example, differs in North American and European countries. All patients with widely metastatic and recurrent diseases experience aggressive and combined therapies with life-threatening toxicities and a low chance of cure. As for OS and EwS, immunotherapy represents a suitable option to increase the patient’s survival, and B7-H3, being expressed by RMS, is an attractive surface antigen for molecular targeted therapy. Notably, a relationship between PAX3-FOXO1 and B7-H3 exists. In ARMS, PAX3-FOXO1 regulates the B7-H3 expression, and the fusion protein and B7-H3 are involved in tumor-promoting pathways such as cell migration and block of differentiation [48].

The efficacy of immunotherapies might be low because most sarcomas are “cold” tumors; although up-regulating immune checkpoints, they express few neoantigens and low MHC-I molecules, and the tumor microenvironment is immunosuppressive and populated by pro-tumorigenic M2 macrophages. On the other hand, B7-H3 is broadly expressed in sarcomas and a recent study showed that B7-H3 CAR-T cells mediate significant antitumor activity in xenograft models, causing regression of solid tumors including osteosarcoma, medulloblastoma, and Ewing sarcoma [49].

Different Phase 1 studies have been designed to target B7-H3 in osteosarcoma, Ewing Sarcoma, and Rhabdomyosarcoma. A Phase 1 study has been completed in the USA that was designed to characterize the safety, tolerability, PK, PD, immunogenicity, and preliminary antitumor activity of enoblituzumab (MGA271), a humanized anti-B7-H3 mAb, in children and young adults with relapsed or refractory malignant solid tumors (ClinicalTrials.gov Identifier: NCT02982941). Three other studies are ongoing that consider the use of B7-H3-CAR-T cells (ClinicalTrials.gov Identifiers: NCT04897321, NCT04864821, and NCT04483778). The latter is enrolling pediatric and young adult patients with relapsed or refractory non-CNS solid tumors evaluating the safety, feasibility, and efficacy of autologous engineered B7-H3-CAR-T cells. CAR-T cells also include tracking proteins, e.g., the cell-surface-localizing truncated forms of EGFR and HER2 (designated EGFRt and HER2tG, respectively). These also serve as safety mechanisms; in the case of overdue toxicity, cetuximab or trastuzumab, targeting EGFRt and HER2tG, respectively, can be used to eliminate the CAR-T cells in patients. Two Arms have been designed in the study; in Arm A patients will receive B7-H3-CAR-T cells only, while in Arm B patients will receive bi-specific B7-H3.CD19-CAR-T cells to improve the persistence of CAR-T cells and B-cell-mediated antigen presentation. An additional trial will evaluate the side effects and the best dose of T cells genetically modified with a 4th generation lentiviral CAR (4SCAR) fused with an inducible apoptotic caspase 9 domain and targeting B7-H3 in refractory and/or recurrent solid tumors (ClinicalTrials.gov Identifier: NCT04432649).

## 4. Acute Leukemia

Acute lymphoblastic leukemia (ALL) represents the most frequent tumor in childhood. High numbers of clinical remission have been obtained with risk-adapted combining chemotherapies, immunotherapies including bi-specific CD19 and CD22 mAbs or CAR-T cells, and allogeneic hematopoietic stem cell transplantation (allo-HSCT) [50]. In addition, a CD19.CD22.CD3 tri-specific mAb has been generated that demonstrated the potential to further overcome tumor escape mechanisms and improve antitumor activity in patient-derived xenograft (PDX) mouse models of B-cell malignancy [51]. Acute myeloid leukemia (AML) [52] has a bimodal distribution affecting adults >65 years and children; it accounts for 15–20% of all pediatric leukemias with a peak incidence in infants aged <1 year. AML shows heterogeneous immunological profiles, both in children and adults. The intensive multi-agent chemotherapy together with the improvement of supportive care increased the 5-year survival rates to up to 70%; a relevant percentage of children (30–40%), however, relapse, and only one third of them survive to adulthood. Immunotherapy approaches such as the immune-checkpoint blockade might be useful in “immune enriched” AMLs. However, most AML shares traits with solid tumors and are fast-growing, aggressive “immune cold” tumors characterized by low mutational load and scarce antigen presentation [53]. Therefore, the development and delivery of new therapeutics remain a priority.

Various studies show that leukemias may express B7-H3 and that, therefore, it could represent a suitable target for immunotherapies. Hu Y. et al., analyzed by flow cytometry the expression of B7-H3 in AML (101 cases) and ALL (33 patients) (range age 1–85) [54]. B7-H3 was detected in 36–50% of M1-M5 AMLs, 54% B-ALL, and 20% ALL. The molecule was preferentially expressed on CD34+ cells and had a prognostic value; indeed, the B7-H3 positive cohort had a shorter overall survival than the B7-H3 negative one [55]. The surface expression of B7-H3 in a subset of AML has been confirmed in an additional study that analyzed blast cells from 111 AML patients. B7-H3 was detected in 27% of patients and its expression was higher in the M3 and M5 subtypes and in cases with mutated NPM1 and wildtype CEBPA. Lichtman et al. [55] showed that B7-H3 is expressed in primary AML blasts from patients with monocytic AML but not in normal bone marrow progenitor populations. They generated B7-H3-CAR-T and demonstrated their antileukemia efficacy in a PDX model, and the lack of significant hematopoietic toxicity in a humanized mouse model. Importantly, B7-H3 acts as an immune-checkpoint ligand suppressing the NK-mediated tumor killing of AML cells [56]. These data support the clinical development of B7-H3-CAR-Ts for the treatment of patients with relapsed/refractory B7-H3 positive AML. To date, two clinical trials only are ongoing that target AML (ClinicalTrials.gov Identifiers: NCT05731219, NCT05722171). Both are recruiting patients of 18 years and older who will be injected with B7-H3-CAR gamma/delta T cells. The primary outcome will be to assess the safety of the treatment and the secondary outcomes to evaluate antitumor activity, overall survival, duration of response, and progression-free survival.

A poorly understood aspect of pediatrics AML is Myeloid sarcomas (MS), extramedullary tumoral masses of myeloid blasts involving different organs [57]. Most often MS occurs contemporarily with intramedullary AML, or as relapse after allo-HSCT, but some MS appears in the absence of the bone marrow disease. MS can affect bone, soft tissues in different districts, skin, CNS, lung and mediastinum, lymph nodes, abdominal cavity, and orbit. Although MS is quite rare in adults, it can affect up to 40% of pediatric patients, particularly infants; tumor cells have peculiar mutational profiles compared to adults and MS is most frequently associated with myelomonocytic (M4) and monocytic (M5) subtypes. Although the MS prognostic relevance is still controversial, its impact on the disease should not be ignored, particularly when considering therapeutic approaches including immunotherapy and allo-HSCT. The development of MS appears to require leukemia mobilization from the BM and tissue homing. Few molecular markers have been described that could be involved in the process including CXCR4, CD56, and some adhesion molecules; moreover, angiogenesis and metalloproteases might contribute to the pathogenesis. Notably, little is known about the MS microenvironment and mechanisms of escape from immune recognition including the expression of immune-checkpoint ligands such as B7-H3. These topics need to be explored to improve diagnosis, better stratify the disease risk, and characterize the efficacy of therapies targeting B7-H3.

## 5. B7-H3 as a Target in Other Pediatric Tumors

A huge number of tumors have been shown to express B7-H3 at the cell surface. In children, besides the above-mentioned pathologies, the targeting of B7-H3 is considered a promising immunotherapeutic approach for the treatment of many other intracranial and extracranial cancers (Table 1) [58]; these also include rare diseases that more frequently lack consolidated and effective therapeutic protocols. Pediatric tumors include those affecting CNS or peripheral nerves, desmoplastic small round cell tumors, soft tissue sarcomas, growing in the abdomen and pelvic area of the body, germ cell cancers originating from reproductive cells and most frequently occurring in the testicles or ovaries, gastric and lung cancers, melanoma and retinoblastoma, an intraocular malignancy with primitive neuroendocrine origins primarily affecting young children. All these tumors have been included in Phase 1 clinical trials (Table 1)

Additional candidates are represented by Wilms tumors, the most common renal tumors in childhood. WT are embryonic tumors with blastemal, epithelial, and stromal components, a triphasic histology that recapitulates the development of the normal kidney. At the molecular level, patients show aberrations in the *WT1* gene and, at low frequency, *CTNNB1* and *TP53* genes [59]. These genes are involved in the Wnt/beta-catenin signaling pathway that has been associated with the regulation of development, cell-cell interactions, cancer progression, and stem cell control [60]. Importantly, the Wnt deregulation is responsible for modulating the inflammatory response in the tumor microenvironment through the regulation of Wnt, RANKL, and FOSL1 pathways; the *FOSL1* gene encodes Fos-like antigen-1 (Fra1) a direct target of IL-17 [7,8]. In WT, as in other tumors, the tumor microenvironment is involved in tumor development and progression; it is also responsible for the epithelial-mesenchymal transition (EMT) and vice versa (MET), which participate in the embryonic development, cancer invasion, metastasis, and drug resistance [61]. Recently, Kendsersky and colleagues analyzed the activity of a drug-conjugated B7-H3-targeting mAb (m276-SL-PBD) in different pediatric solid tumors including WT, and showed significant antitumor activity in patients derived (PDX) and cell line-derived (CDX) xenograft models [25]. Three clinical trials are ongoing that target B7-H3 in WT patients; one is completed and utilized a mAb (ClinicalTrials.gov Identifier: NCT02982941), and the others are still recruiting and will use as therapeutics B7-H3-CAR-T cells (ClinicalTrials.gov Identifiers: NCT0489732, NCT04483778) (Table 1)

## 6. Therapeutics Targeting B7-H3

As mentioned above different therapeutics can be used to target B7-H3. Naked chimeric, humanized and, soon, 100% human mAbs, can be used to block the interaction between B7-H3 and its inhibitory receptor reactivating or enhancing the antitumor function of NK cells, T cells, and macrophages (Figure 1). These can be used alone or in combination with mAbs blocking immunomodulatory cytokines present in the tumor microenvironment and impairing the function of immune effectors; for example, galunisertib, an inhibitor of TGFβR1 has been shown to increase the efficacy of the anti-GD2 mAb and restore the NK cytolytic activity against neuroblastoma [62,63]. Bi-specific and tri-specific mAbs have been generated that more tightly link immune effectors to tumor cells. Bi-specific mAbs were built to engage B7-H3 and surface molecules expressed by T cells (bi-specific T cell engager, BiTE), these include CD3 and 4-1BB (CD137) that is expressed on PD-1+ Tim-3+ “terminally differentiated” cells [64]. BiTEs induced, in vitro and in vivo in human tumor mice models, T-cell activation, proliferation, potent antitumor activity, and memory formation, whereas undesired cytokine release was reduced. Interestingly, the B7-H3.4-1BB. BiTE acted synergistically with the PD-1 blockade. Valera D.A. and colleagues generated a tri-specific killer engager (TriKE) assembling an anti-B7-H3 scFv, a camelid anti-CD16 antibody fragment, and IL-15 [65]. TriKE enhanced the NK cell function, proliferation, and cytotoxicity against various types of human cancer cell lines in vitro and tumor-bearing NSG mice.

Recently, tetra-specific molecules (termed natural killer cell engager, NKCE) have been generated engaging a TAA, two activating NK receptors (NKp46 and CD16a), and carrying a point mutated variant of the IL-2 stimulatory cytokine (IL-2v) [66] (Figure 1). IL-2v interacts with the low-affinity receptor (IL-2Rbeta/gamma, CD122-CD132) on memory CD8 T cells and NK cells, but not with the trimeric receptor, which includes IL-2Ralpha (CD25), predominantly found on Treg; this avoids the unwanted stimulation of cells with potent immunomodulatory properties. NKCE targeting CD123 (SAR’579/IPH6101) showed potent antitumor activity in vitro, against primary AML blasts, and, in vivo, in a mouse CD123+ tumor model. Moreover, in nonhuman primates, it had prolonged pharmaco-dynamic effects, and depleted CD123+ cells without signs of toxicity and cytokine release syndrome. Results provided the rationale for the design of clinical trials with CD123-NKCE in AML patients (ClinicalTrials.gov Identifier: NCT05086315) and suggest the possibility to generate NKCE targeting additional TAA. In June 2023 SAR’579 received FDA fast-track designation in the US for the treatment of hematological malignancies. In December 2022, Sanofi and Innate Pharma announced an expansion of their collaboration, with Sanofi licensing an NKCE program targeting B7-H3.

CAR effectors also rapidly evolved (Figure 2). More and more efficient stimulatory signals have been added to the constructs, and CAR cells carry safety suicide genes such as the inducible caspase 9 (iCasp9), which can be activated on demand by administering the dimerizing agent AP1903 (rimiducid). Moreover, as for mAbs, also bi-specific CARs have been generated, that target two TAA, or a TAA and a molecule expressed by immune effectors. Recently, using the synthetic Notch design, a bi-specific GD2.B7-H3-CAR-T was generated characterized by inducible CAR expression; the expression of the B7-H3-CAR depends on the initiation of a transactivating signal mediated by the GD2-CAR [67]. The inducible CAR-T cells had a high metabolic fitness and eradicated GD2+ B7-H3+ tumors in NSG mice inoculated intravenously with different human tumor cell lines. The use of inducible CARs is interesting and could reduce the systemic side effects. On the other hand, their efficacy may be lost in tumors decreasing or abolishing the expression of the main TAA, as occurs for GD2-negative neuroblastoma variants [23].

A further possibility is to genetically engineer NK cells, thus combining their natural antitumor cytotoxicity, which is mediated by various activating receptors recognizing ligands on tumor cells [7], with the CAR specificity for selected TAA (Figure 2). For example, CAR-NK cells have been generated that target CD123, an antigen highly expressed by AML. The scFv derived from a murine anti-CD23 mAb was linked via the human CD8 hinge-transmembrane domain to 4-1BB (CD137) and CD3-zeta costimulatory domains, and a peptide derived from the human CD34 was added as a trackable marker [68]. CD123-CAR-NK cells had antileukemia activity against AML cell lines and primary blasts from pediatric patients both in vitro and in human AML-bearing immune-deficient mice. Importantly, the toxicity of CD123-CAR-NK was significantly lower as compared with that of CD123-CAR-T cells.

## 7. Conclusions

Immunotherapy turned out to be a promising approach in both hematological malignancies and solid tumors. B7-H3 is a good TAA being expressed by different tumors, in different disease stages, and by cells populating the tumor microenvironment such as tumor-associated endothelial cells, macrophages, and fibroblasts. As described above, a huge variety of therapeutic tools are or will be available to restore immune responses and/or target tumor cells; engineered effectors have rapidly evolved together with the comprehension that combining approaches have major chances of success.

Overall, published data strongly encourage the application of B7-H3 targeted immunotherapies in different pediatric tumors. The translation of preclinical data to the clinical application, however, will be further supported by the set-up of preclinical models better reconstituting the tumor microenvironment, which often is highly immunosuppressive. Preclinical human tumor mouse models, including PDX, are challenged by the tumor evolution during the engraftment, including the in vivo selection of tumor variants, the lack of the host immunological response, and the presence of xenogeneic tumor microenvironment. Studies are ongoing to develop alternative strategies for testing immunotherapies and better personalizing the therapeutic approaches. Promising models include the development of humanized mice carrying human hematopoietic cells, which generate different immune cells including macrophages and dendritic cells [69], and the generation of three-dimensional (3D) cell cultures or organs-on-chip in vitro models, which allow cells to grow in a spatial organization more like in vivo tissues, possibly preserving the tumor microenvironment [70,71].

## Figures and Tables

**Figure 1 cancers-15-03279-f001:**
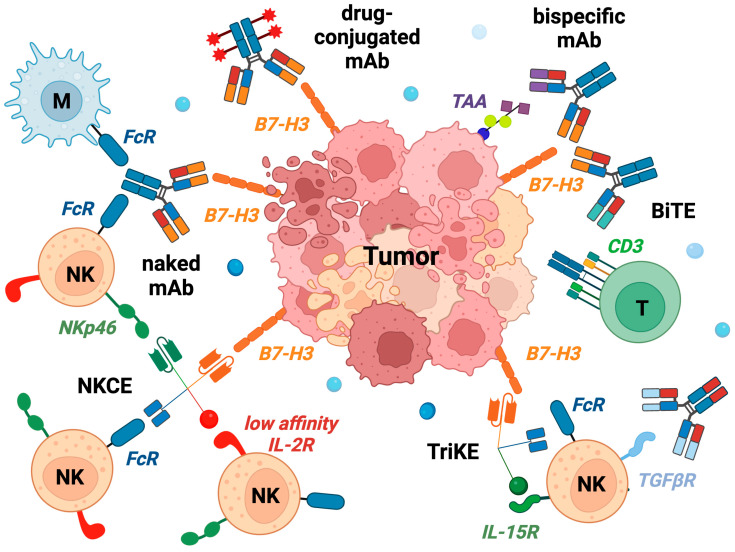
Overview of the mAbs and engineered engagers targeting B7-H3. NK, natural Killer cells; T, T cells; M, macrophages; TAA, tumor-associated antigen; FcR, Fc receptor; BITE, bi-specific T-cell engager; TriKE, tri-specific killer engager; NKCE, natural killer cell engager; IL-2R, IL-15R, and TGFbetaR, receptors for IL-2, IL-15, and TGFbeta cytokines, respectively (Created by BioRender.com (URL accessed on 16 June 2023)).

**Figure 2 cancers-15-03279-f002:**
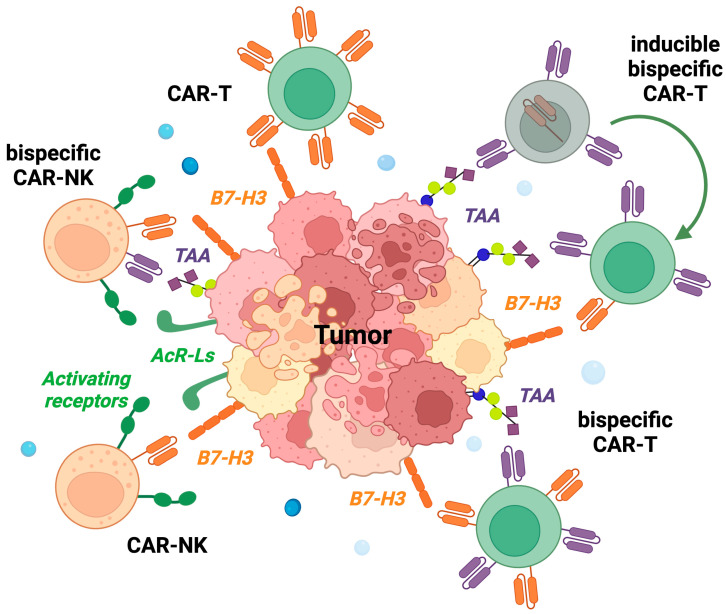
Overview of the CAR-engineered effectors targeting B7-H3. NK, natural Killer cells; T, T cells; TAA, tumor-associated antigen; AcR-Ls activating receptor ligands. Created by BioRender.com (URL accessed on 16 June 2023).

**Table 1 cancers-15-03279-t001:** Currently active/recruiting/completed pediatric immunotherapy clinical trials involving B7-H3. Those involving neuroblastoma patients are summarized by Pulido R. et al. [27].

NCT Number	Title	Status	Study Results	Conditions	Interventions	Age	Phases	Locations
ClinicalTrials.gov Identifier: NCT05063357	131I-omburtamab Delivered by Convection-Enhanced Delivery in Patients With Diffuse Intrinsic Pontine Glioma	Not yet recruiting	No Results Available	Diffuse Intrinsic Pontine Glioma	Drug: 131I-Omburtamab|Device: Convention Enhanced Delivery	3 Years to 21 Years (Child, Adult)	Phase 1	Y-mAbs Therapeutics, Labcorp Corporation of America Holdings, Inc. Invicro, Burlington, North Carolina
ClinicalTrials.gov Identifier: NCT04167618	177Lu-DTPA-Omburtamab Radioimmunotherapy for Recurrent or Refractory Medulloblastoma	Terminated	No Results Available	Medulloblastoma, Childhood	Drug: 177Lu-DTPA-omburtamab	3 Years to 19 Years (Child, Adult)	Phase 1| Phase 2	Mayo Clinic, Rochester, MN, United States| Memorial Sloan Kettering Cancer Center, New York, NY, United States|Doernbecher Children’s Hospital, Portland, OR, United States|M.D. Anderson Cancer Center, Houston, TX, United States|Rigshospitalet, Børneonkologisk afsnit, Copenhagen, Denmark|Princess Máxima, Utrecht, Netherlands|Hospital Universitari Vall d’Hebron, Barcelona, Spain|Hospital Sant Joan de Déu de Barcelona, Barcelona, Spain|The Royal Marsden Hospital, London, United Kingdom|Great North Children’s Hospital, Newcastle, United Kingdom
ClinicalTrials.gov Identifier: NCT03275402	131I-omburtamab Radioimmunotherapy for Neuroblastoma Central Nervous System/Leptomeningeal Metastases	Active, not recruiting	No Results Available	Neuroblastoma|CNS Metastases| Leptomeningeal Metastases	Biological: 131I-omburtamab	up to 18 Years (Child, Adult)	Phase 2| Phase 3	Children’s Hospital Los Angeles, Los Angeles, CA, United States|Riley Hospital for Children, Indianapolis, IN, United States|Memorial Sloan Kettering Cancer Center, New York, NY, United States|Nationwide Children’s Hospital, Columbus, OH, United States|M.D. Anderson Cancer Center, Houston, TX, United States|Rigshospitalet, København, Denmark| Department of Pediatric Oncology Fukushima Medical University Hospital, Fukushima City, Japan|Hospital Sant Joan de Déu, Barcelona, Spain
ClinicalTrials.gov Identifier: NCT04743661	131I-Omburtamab, in Recurrent Medulloblastoma and Ependymoma	Active, not recruiting	No Results Available	Recurrent Medulloblastoma| Recurrent Ependymoma	Drug: Irinotecan|Drug: Temozolomide|Drug: Bevacizumab|Drug: Omburtamab I-131|Drug: Liothyronine|Drug: SSKI|Drug: Dexamethasone|Drug: Antipyretic|Drug: Antihistamine|Drug: anti-emetics	up to 21 Years (Child, Adult)	Phase 2	Children’s Hospital Los Angeles, Los Angeles, CA, United States|Memorial Sloan Kettering Cancer Center, New York, NY, United States|Cincinnati Children’s Hospital Medical Center, Cincinnati, OH, United States
ClinicalTrials.gov Identifier: NCT01502917	Convection-Enhanced Delivery of 124I-Omburtamab for Patients With Non-Progressive Diffuse Pontine Gliomas Previously Treated With External Beam Radiation Therapy	Completed	No Results Available	Brain Cancer|Brain Stem Glioma	Drug: Radioactive iodine-labeled monoclonal antibody omburtamab|Radiation: External Beam Radiotherapy	2 Years to 21 Years (Child, Adult)	Phase 1	Weill Medical College of Cornell University, New York, NY, United States|Memorial Sloan Kettering Cancer Center, New York, NY, United States
ClinicalTrials.gov Identifier: NCT05064306	131I-omburtamab for the Treatment of Central Nervous System/Leptomeningeal Neoplasms in Children and Young Adults	Available	No Results Available	Central Nervous System/ Leptomeningeal Neoplasms	Drug: 131I-omburtamab	Child, Adult, Older Adult		Memorial Sloan Kettering Cancer Center, New York, NY, United States
ClinicalTrials.gov Identifier: NCT04022213	A Study of the Drug I131-Omburtamab in People With Desmoplastic Small Round Cell Tumors and Other Solid Tumors in the Peritoneum	Recruiting	No Results Available	Desmoplastic Small Round Cell Tumor|Peritoneal Cancer| Peritoneal Carcinoma	Drug: 131 I-omburtamab|Radiation: WAP-IMRT	1 Year and older (Child, Adult, Older Adult)	Phase 2	Memorial Sloan Kettering Cancer Center, New York, NY, United States
ClinicalTrials.gov Identifier: NCT00089245	Radiolabeled Monoclonal Antibody Therapy in Treating Patients With Refractory, Recurrent, or Advanced CNS or Leptomeningeal Cancer	Active, not recruiting	No Results Available	Brain and Central Nervous System Tumors|Neuroblastoma| Sarcoma	Drug: Iodine I 131 MOAB 8H9	Child, Adult, Older Adult	Phase 1	Memorial Sloan Kettering Cancer Center, New York, NY, United States
ClinicalTrials.gov Identifier: NCT04185038	Study of B7-H3-Specific CAR-T Cell Locoregional Immunotherapy for Diffuse Intrinsic Pontine Glioma/Diffuse Midline Glioma and Recurrent or Refractory Pediatric Central Nervous System Tumors	Recruiting	No Results Available	Central Nervous System Tumor|Diffuse Intrinsic Pontine Glioma|Diffuse Midline Glioma|Ependymoma| Medulloblastoma, Childhood|Germ Cell Tumor|Atypical Teratoid/Rhabdoid Tumor| Primitive Neuroectodermal Tumor|Choroid Plexus Carcinoma|Pineoblastoma, Childhood|Glioma	Biological: SCRI-CARB7H3(s); B7H3-specific chimeric antigen receptor (CAR) T cel	1 Year to 26 Years (Child, Adult)	Phase 1	Seattle Children’s Hospital Seattle, WA, United State
ClinicalTrials.gov Identifier: NCT05768880	Study of B7-H3, EGFR806, HER2, And IL13-Zetakine (Quad) CAR-T Cell Locoregional Immunotherapy For Pediatric Diffuse Intrinsic Pontine Glioma, Diffuse Midline Glioma, And Recurrent Or Refractory Central Nervous System Tumors	Not yet recruiting	No Results Available	Diffuse Intrinsic Pontine Glioma|Diffuse Midline Glioma|Recurrent CNS Tumor, Adult|Recurrent, CNS Tumor, Childhood|Refractory Primary Malignant Central Nervous System Neoplasm	Biological: SC-CAR4BRAIN	1 Year to 26 Years (Child, Adult)	Phase 1	Seattle Children’s Hospital Seattle, WA, United States
ClinicalTrials.gov Identifier: NCT02982941	Enoblituzumab (MGA271) in Children With B7-H3-expressing Solid Tumors	Completed	No Results Available	Neuroblastoma| Rhabdomyosarcoma| Osteosarcoma| Ewing Sarcoma|Wilms Tumor|Desmoplastic Small Round Cell Tumor	Drug: Enoblituzumab	1 Year to 35 Years (Child, Adult)	Phase 1	Lucile Packard Children’s Hospital, Stanford, Palo Alto, California; National Cancer Institute, Center for Cancer Research, Bethesda, Maryland; Children’s Hospital of Philadelphia, Philadelphia, Pennsylvania; Texas Children’s Hospital Houston, Texas; Seattle Children’s, Seattle, Washington; University of Wisconsin, American Family Children’s Hospital, Madison, Wisconsin
ClinicalTrials.gov Identifier: NCT04897321	B7-H3-Specific Chimeric Antigen Receptor Autologous T-Cell Therapy for Pediatric Patients With Solid Tumors (3CAR)	Recruiting	No Results Available	Pediatric Solid Tumor| Osteosarcoma| Rhabdomyosarcoma| Neuroblastoma|Ewing Sarcoma|Wilms Tumor| Adrenocortical Cancer| Desmoplastic Small Round Cell Tumor|Germ Cell Cancer|Rhabdoid Tumor|Clear Cell Sarcoma|Hepatoblastoma| Melanoma|Carcinoma|Malignant Peripheral Nerve Sheath Tumors|Soft Tissue Sarcoma	Drug: Fludarabine Drug: Cyclophosphamide Drug: MESNA| Drug: B7-H3 CAR-T cells	up to 21 Years (Child, Adult)	Phase 1	St. Jude Children’s Research Hospital, Memphis, TN, United States
ClinicalTrials.gov Identifier: NCT04864821	Clinical Study of CD276 Targeted Autologous Chimeric Antigen Receptor T-Cell Infusion in Patients With CD276 Positive Advanced Solid Tumor	Not yet recruiting	No Results Available	Osteosarcoma|Neuroblastoma| Gastric Cancer|Lung Cancer	Drug: Targeting CD276 CAR-T cells	1 Year to 70 Years (Child, Adult, Older Adult)	Early Phase 1	PersonGen BioTherapeutics (Suzhou) Co., Ltd., The First Affiliated Hospital of Zhengzhou University
ClinicalTrials.gov Identifier: NCT04483778	B7H3 CAR-T Cell Immunotherapy for Recurrent/Refractory Solid Tumors in Children and Young Adults	Recruiting	No Results Available	Pediatric Solid Tumor|Germ Cell Tumor|Retinoblastoma| Hepatoblastoma|Wilms Tumor|Rhabdoid Tumor| Carcinoma|Osteosarcoma| Ewing Sarcoma| Rhabdomyosarcoma| Synovial Sarcoma|Clear Cell Sarcoma|Malignant Peripheral Nerve Sheath Tumors|Desmoplastic Small Round Cell Tumor|Soft Tissue Sarcoma| Neuroblastoma|Melanoma	Biological: second generation 4-1BBz B7H3-EGFRt-DHFR Biological: second generation 4-1BBz B7H3-EGFRt-DHFR(selected) and a second generation 4-1BBz CD19-Her2tG	0 Years to 26 Years (Child, Adult)	Phase 1	Seattle Children’s Hospital, Seattle, WA, United States
ClinicalTrials.gov Identifier: NCT04432649	Targeting CD276 (B7-H3) Positive Solid Tumors by 4SCAR-276	Recruiting	No Results Available	Solid Tumor	Biological: 4SCAR-276	1 Year to 75 Years (Child, Adult, Older Adult)	Phase 1| Phase 2	Shenzhen Children’s Hospital, Shenzhen Geno-immune Medical Institute, Sun Yat-Sen University, Shenzhen, Guangdong, China
ClinicalTrials.gov Identifier: NCT05731219	UTAA06 Injection in the Treatment of Relapsed/Refractory Acute Myeloid Leukemia	Recruiting	No Results Available	Relapsed/Refractory Acute Myeloid Leukemia	Biological: B7-H3 target, CAR gene-modified gdT cell injection	18 Years and older (Adult, Older Adult)	Phase 1	The First Affiliated Hospital, Zhejiang University School of Medicine Hangzhou, Zhejiang, China
ClinicalTrials.gov Identifier: NCT05722171	Clinical Study of UTAA06 Injection in the Treatment of Relapsed/Refractory Acute Myeloid Leukemia	Recruiting	No Results Available	Relapsed/Refractory Acute Myeloid Leukemia	Biological: gdT cell injection targeting B7-H3 chimeric antigen receptor	18 Years and older (Adult, Older Adult)	Early Phase 1	PersonGen Anke Cellular Therapeutics Co., Ltd. Hefei, Anhui, China

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
