# Peer review of "B7-H3 in Pediatric Tumors: Far beyond Neuroblastoma"

_cancers, 2023, doi:10.3390/cancers15133279_

Round 1

Reviewer 1 Report

In this manuscript Bottino et al seek to review the role of B7-H3 in pediatric tumors and how it may be taken advantage of to develop more tolerable therapies.  In particular the focus is children, who have significantly longer durations to experience chemotherapy, radiation, and surgical-induced side effects compared to the elderly, in addition to more treatment related side effects in the future such as treatment-related malignancies and cardiac disease.  Overall, the authors cover a significant amount of information.  Unfortunately, it is often disorganized, taking away from the point of emphasis for that particular section.  With some heavy editing and clarification, this could be a very nice review.  Below are some examples of what needs to be improved, but it is not all-inclusive as these problems were noted throughout each section. 

1. The last sentence in the simple summary is quite underwhelming.  Offer a it more here as to the hopes of B7H3 targeting agents. 

2. Line 23, "life-treating" should be "life-limiting" or some other equivalent.  

3. The introduction can be quite vague at times regarding results and findings.  Please add more details rather than saying for example: ...demonstrated the immunoregulatory role of B7-H3... as in lines 49-50.  Was it positive or negative regulatory? Several other occurrences like this throughout the intro that leave the reader confused as to the point being made and wanting additional information.  

4. Lines 60-62.  Both TLT2 and IL-13R alpha have been proposed as possible receptors for B7-H3, although little has been done to confirm these. 

5. Line 72-74, consider mentioning the work by Orme et al on soluble B7H3 and its effects on solid tumors.  More detail on this (and in general would make this a more useful review to the readers).  

6.  Lines 80-87 don't seem to fit the rest of the paragraph.  Keep with intro material and hold off on neuroblastoma therapies until that section.  Expand upon the tumor microenvironment a bit and how B7H3 affects it. 

7. Table 1, consider adding recruiting institution.  

8. Line 97, "first" common should be "most" common. 

9. Lines 115-134 are out of place.  Consider a separate section on modalities to target B7H3 in the intro.  

10. Lines 135-143 really gloss over a lot of work with I131 and Lu177.  Please expand on this with details and results from the trials.  This supports your review that B7H3 targeting is important and possibly effective and better tolerated.  

11. Lines 144-153.  I believe the EGFRt is actually a kill switch/target.  Please confirm and update the description of this.  You mention this use later in the review, but it is not clear at all that the EGFRt 1. is an antigen targeting region or 2. is actually a kill target/regulatory control.  This is a huge part of this CAR-T and needs to be included.  

12. Numerous symbols inappropriately converted throughout (Spiral noted in line 157 among others). 

13. The sarcoma section has a lot on the genetics of sarcoma, but is missing survival statistics for EwS and RMS, which are noted for OS.  Please be consistent in the data you are presenting.  if you present for one, then please include in the other sections as well for clarity.  

14.  There is very little reference to B7H3 in teh sarcoma section (#3).  Please expand on this more. 

15. Lines 221-231, while this is interesting, I do not understand why it is here.  There is no reference to B7-H3 in the entire paragraph. 

16. Lines 247-249 now reference a safety/kill switch for the CAR products from Fred Hutch/Seattle Children's. 

17.  Line 268, we typically think of AML having a "bimodal" distribution affecting <1 year and >65.  Please correct/update. 

18. Lines 330-359 belong in the body of the manuscript, not the conclusion.  They are adding additional information rather than summarizing and pointing towards future directions/challenges. 

19.  Bispecifics and other NK cell therapies targeting B7H3 have not been mentioned.  There are several with possible pediatric targets that have not been mentioned that focus on B7-H3.  Consider expanding on this.  There are also several trials from China registered on clinicaltrials.gov.  Please consider updating to include these. 

20.  consider a "modalities of targeting" section including mAb, antibody drug conjugates, immune engagers, and engineered cells (T and NK cells). This may make things a bit clearer and easier to focus on each individual tumor.  Consider mentioning rare childhood tumors as well, these are very good targets for B7H3 therapies.  

21.  A little underwhelming with one table and one figure.  Please consider adding another figure or two at least to increase the impact and usefulness for the audience. 

Overall the English is pretty good.  In some places the language is a bit vague or colloquial for a scientific journal.  Cases need to be matched (singular subject with singular conjugated verb).  There are many discrepancies of this throughout the manuscript.  

Author Response

Dear Editors and Reviewers,

We are submitting the revised version of our Review entitled “B7-H3 in Pediatric Tumors: Far Beyond Neuroblastoma” modified according to the Reviewers’ comments and suggestions. Find below the point-by-point reply.

Changes are highlighted in yellow.

We hope that the new version of the manuscript deserves consideration for publication in Cancers.

Yours sincerely,

Cristina Bottino

REVIEWER 1

  1. The last sentence in the simple summary is quite underwhelming. Offer a it more here as to the hopes of B7H3 targeting agents.

We strongly believe in the anti-tumor efficacy of B7-H3 targeting in cancer patients. However, in our opinion, caution is always best; for example, the PD-1 blockade showed astonishing successes but also unexpected failures.

  1. Line 23, "life-treating" should be "life-limiting" or some other equivalent.

The typo has been corrected.

  1. The introduction can be quite vague at times regarding results and findings. Please add more details rather than saying for example: ...demonstrated the immunoregulatory role of B7-H3... as in lines 49-50. Was it positive or negative regulatory? Several other occurrences like this throughout the intro that leave the reader confused as to the point being made and wanting additional information.

The neuroblastoma has been described in the introduction because, in our opinion, is the “B7-H3 starting point”. The function of B7-H3, the preclinical data on B7-H3 targeting, and the ongoing clinical trials are out of the scope of the present review and have been nicely summarized in different (cited) reviews.

  1. Lines 60-62. Both TLT2 and IL-13R alpha have been proposed as possible receptors for B7-H3, although little has been done to confirm these.

To our knowledge, B7-H3 is still an orphan ligand in humans and TLT2 is the only receptor identified so far in mice.

  1. Line 72-74, consider mentioning the work by Orme et al on soluble B7H3 and its effects on solid tumors. More detail on this (and in general would make this a more useful review to the readers).

We apologize to the Referee; we were unable to find a paper published by Orme on soluble B7-H3.

Regarding the possibility to add more detail on the soluble form, this is out of the scope of the review: data are nicely summarized in most cited reviews.

  1. Lines 80-87 don't seem to fit the rest of the paragraph. Keep with intro material and hold off on neuroblastoma therapies until that section.  Expand upon the tumor microenvironment a bit and how B7H3 affects it.

In this part of the introduction, we just mention that the microenvironment is important for cancers development and that different ongoing clinical trials are ongoing. Both issues are nicely summarized in the cited reviews.

  1. Table 1, consider adding recruiting institution.

The recruiting institutions have been added in Table 1.

  1. Line 97, "first" common should be "most" common.

The sentence has been modified.

  1. Lines 135-143 really gloss over a lot of work with I131 and Lu177. Please expand on this with details and results from the trials. This supports your review that B7H3 targeting is important and possibly effective and better tolerated. 

To highlight these studies, the related clinical trials have been listed in Table 1.

  1. Lines 115-134 are out of place. Consider a separate section on modalities to target B7H3 in the intro.
  2. Lines 330-359 belong in the body of the manuscript, not the conclusion. They are adding additional information rather than summarizing and pointing towards future directions/challenges.
  3. Bispecifics and other NK cell therapies targeting B7H3 have not been mentioned. There are several with possible pediatric targets that have not been mentioned that focus on B7-H3.  Consider expanding on this.  There are also several trials from China registered on clinicaltrials.gov.  Please consider updating to include these.
  4. consider a "modalities of targeting" section including mAb, antibody drug conjugates, immune engagers, and engineered cells (T and NK cells). This may make things a bit clearer and easier to focus on each individual tumor. Consider mentioning rare childhood tumors as well, these are very good targets for B7H3 therapies.

As suggested, we added a separate section on tools targeting B7-H3 and, as requested by Referee 3, an additional section mentioning other tumors.

Regarding the trials, there are many worldwide interesting trials. However, the majority involve adults and are nicely summarized in other reviews.

  1. Lines 144-153. I believe the EGFRt is actually a kill switch/target. Please confirm and update the description of this.  You mention this use later in the review, but it is not clear at all that the EGFRt 1. is an antigen targeting region or 2. is actually a kill target/regulatory control.  This is a huge part of this CAR-T and needs to be included.
  2. Lines 247-249 now reference a safety/kill switch for the CAR products from Fred Hutch/Seattle Children's.

The sentence has been modified according to the Referee’s suggestion.

  1. Numerous symbols inappropriately converted throughout (Spiral noted in line 157 among others).

The typos have been corrected.

  1. The sarcoma section has a lot on the genetics of sarcoma, but is missing survival statistics for EwS and RMS, which are noted for OS. Please be consistent in the data you are presenting. if you present for one, then please include in the other sections as well for clarity
  2. There is very little reference to B7H3 in teh sarcoma section (#3). Please expand on this more.

The survival statistics are out of the scope of this review and have been removed. We focused on genetics because of the low mutation rate of these tumors.

The section includes additional references.

  1. Lines 221-231, while this is interesting, I do not understand why it is here. There is no reference to B7-H3 in the entire paragraph.

The paragraph has been deleted.

  1. Line 268, we typically think of AML having a "bimodal" distribution affecting <1 year and >65. Please correct/update.

The sentence has been corrected.

  1. A little underwhelming with one table and one figure. Please consider adding another figure or two at least to increase the impact and usefulness for the audience.

Therapeutics targeting B7-H3 are now illustrated in figures fig. 1 and 2.

Comments on the Quality of English Language

Overall the English is pretty good.  In some places the language is a bit vague or colloquial for a scientific journal.  Cases need to be matched (singular subject with singular conjugated verb).  There are many discrepancies of this throughout the manuscript. 

The typos have been corrected.

REVIEWER 2

  1. There are numerous typos and errors in the English language. For instance, "life treating side effects" should be corrected to "life threatening," "T cel" should be "T cell," "On the other hands" should be "On the other hand," and "that of ERMS shows two picks" should be "that of ERMS shows two peaks."

The typos have been corrected.

  1. In section 1, Introduction, paragraph 3, the author mentions that B7-H3 is also expressed on macrophages (TAM) and cancer-associated fibroblast (CAF). However, the authors do not address any advancements in using B7-H3 as a therapeutic target to overcome the immunosuppressive barriers of TAM or CAF in solid tumor microenvironments, such as glioma, sarcomas, or Myeloid sarcomas.

The possibility that targeting B7-H3 could also modify the microenvironment is appealing. However, little is known about this issue. The topic is summarized by different cited reviews.

  1. The objective of the review is to discuss "B7-H3 in Pediatric Tumors: Far beyond Neuroblastoma" and emphasize the significance of utilizing B7-H3 as an immune-therapeutic target. However, the authors dedicate a significant amount of time to discussing each cancer type. For example, in section 2, paragraphs 1 to 3 focus on glioma; in section 3, paragraphs 1-3 discuss sarcoma; in section 4, paragraph 1 covers another cancer type. It would be more appropriate to focus on the hurdles or challenges related to the current therapy for these cancers.

The review aims at summarizing data on neuroblastoma (with updated references) and present recent data on other pediatric aggressive cancers; in our opinion, a brief description of each pathology will be appreciated by general readers.  

  1. Furthermore, these sections lack appropriate references, contain poorly written passages, and, in some cases, present incorrect statements. For example, the statement "Unlike OS and EwS, less is known about the genetic alterations associated with RMS" is incorrect, as there are several publications on the genomics and epigenetics of rhabdomyosarcoma. The authors should also discuss and present the expression of B7-H3 mRNA and protein across these cancers, as this data is publicly available in numerous databases.

We apologize for the misleading sentence; it has been deleted and we added additional references.

Regarding the B7-H3 expression, this molecule is expressed on the cell surface of tumors and represents a suitable target; on the contrary, different studies show that the mRNA expression is not a reliable criterium since it can be detected in cells that do not express B7-H3 at the cell surface, including healthy tissues.

  1. In section 2, paragraph 4, it is necessary to clarify why the FDA halted the use of I131-omburtamab (8h9) in children with CNS or leptomeningeal metastases from neuroblastoma.

Omissis…The ODAC’s vote was based on findings from the phase 1 study 03-133 (NCT00089245), which assessed omburtamab in a population with relapsed, recurrent, or advanced CNS or leptomeningeal cancer, as well as the phase 1/2 study 101 (NCT03275402) examining omburtamab in neuroblastoma with CNS or leptomeningeal metastases. Members of ODAC concluded that there was not enough evidence to support omburtamab’s overall survival benefit in this patient population”…omissis

  1. This manuscript should incorporate the advancements in current pre-clinical research on B7-H3 targeted therapy, as well as the advantages and disadvantages of different generations of B7-H3 CAR, bispecific CARs and armored CARs, to enhance the potency of B7-H3 CAR T therapy.
  2. The authors should discuss the potential on target off tumor toxicities of B7-H3 targeted therapies, due to expression in normal tissues/organs

Preclinical data are promising; however, mice are not humans and to date, we have no results on the efficacy and toxicity in pediatric patients since clinical trials are ongoing.

B7-H3 does not appear to be expressed in normal tissue; this is mentioned in the introduction.

  1. The detailed discussion of EGFRt does not align well with the scope of this review.

It has been described since it is a safety/kill switch for the CAR products (see Q11 and 16, referee 1).

  1. In the conclusion, paragraph 2, instead of introducing CD122-CD132 bi-specific mAbs or CD123 NKCE, the authors should summarize the research advances of B7-H3 directed immune therapies.

As requested by Reviewer 1, the conclusion section has been modified. Most data are now presented in the new section on B7-H3 targeting.

    There are numerous typos and errors in the English language. For instance, "life treating side effects" should be corrected to "life threatening," "T cel" should be "T cell," "On the other hands" should be "On the other hand," and "that of ERMS shows two picks" should be "that of ERMS shows two peaks."

The typos have been corrected.

REVIEWER 3

  1. The first paragraph in the introduction should include indication of other antibodies/studies that identified B7-H3 expressed on tumor cells, including antibody 376.96

The mAb has been cited in the new paragraph on B7-H3 targeting (requested by referee 1)

  1. Authors should include “paediatric” in the sentence “this review aims at discussing B7-H3 as a possible therapeutic target in tumors” in the last paragraph of the introduction (“…target in paediatric tumors”)

It has been added.

  1. Authors must be consistent and choose uniform nomenclature throughout the manuscript (“B7-H3 CAR-T”, “B7-H3.CAR-T” or “B7-H3-CAR T”). It is suggested not to use the form “B7-H3.CAR-T” (used for most of CAR-T indications) because of the ambiguity with the dot which gets confused with the orthographic dot.

B7-H3-CAR-T was chosen as uniform nomenclature.

  1. In the conclusion section, the authors start with mentioning Wilms tumor, which was not mentioned previous in the text. Could authors make a paragraph prior to conclusion with “other paediatric tumors”, which could include more paediatric cancers forms that as included in Table 1? In this novel section, authors could include relevant references, including the recent review PMID: 37110590

As requested, we added a novel section entitled “B7-H3 as a target in other pediatric tumors”.  The review has been cited.

  1. Authors should include a more informative figure legend to Figure 1, explaining the different forms of targeting B7-H3, as well as explaining the abbreviations used the Figure.

As requested by Referee 1, data are depicted in two figures. The legends have been modified.

  1. Title of Table 1 is “Currently active/recruiting”, but table also includes “completed” studies. This should be corrected.

It has been added.

  1. To better put the readers in the appropriate context, authors could refer in the Introduction section to more comprehensive reviews on B7-H3 in cancer (not restricted to paediatric), such as PMID: 31099317 and 36859240

References have been added.

  1. Other small corrections/typos:

-Line 18: the abbreviation “4Ig” stands for “4 immunoglobulin”. Please amend this.

- Line 28: the abbreviation mAb stands for “monoclonal antibodies”. Please amend this.

- Line 44: please write “4 immunoglobulin domains” instead of “4 Ig domains”.

4Ig and mAbs are commonly used acronyms. These were explained when used for the first time.

Line 58: Please explain “B7-H3R/B7-H3 axis”

The sentence has been changed.

Line 72: Please amend abbreviation to “soluble B7-H3 (sB7-H3)”, as this is used later on.

Line 117: the abbreviation “TAA” stands for “Tumor-associated antigens”. Please amend this.

Line 120: The abbreviation “PD-1/PD-Ls”. “Ls” is ambiguous. Please amend this.

Line 142: Please remove abbreviation “(ODAC)” as this is not further used.

Line 148: delete “a” from “is a non-immunogenic,…”

Line 229: Please correct “clear” to “partial”.

Line 268: please remove “,” after “[48]”

Line 281: please add punctuation after “[50]”

Line 289: Please correct “Y. Hu” to “Hu et al.”

Line 295: it is suggested to delete “only”

Line 297: please amend the symbols. They also need amendment in lines 297, 331, 3336, 338

Line 338: change “unwonted” by “unwanted”

Line 370: it is suggested to change “realization” by “generation”

Line 228: Please correct “TNF- , IFN- , and IL-1” to “TNF-α, IFN-γ, and IL1-β”.

done

All changes were made according to the Referee’s requests.

Line 224: the expression “In the late 800’…” is ambiguous. Please amend this.

As requested by Referee 1, the paragraph has been deleted.

Table includes nomenclature “CD276”. Please amend this.

The CD of the B7-H3 molecule is now mentioned in the introduction.

Reviewer 2 Report

  1. This manuscript aimed to address the significance and advancements in utilizing B7-H3 as an immune-therapeutic target in cancer. However, the paper has several notable deficiencies.

    1.     There are numerous typos and errors in the English language. For instance, "life treating side effects" should be corrected to "life threatening," "T cel" should be "T cell," "On the other hands" should be "On the other hand," and "that of ERMS shows two picks" should be "that of ERMS shows two peaks."

    2.     In section 1, Introduction, paragraph 3, the author mentions that B7-H3 is also expressed on macrophages (TAM) and cancer-associated fibroblast (CAF). However, the authors do not address any advancements in using B7-H3 as a therapeutic target to overcome the immunosuppressive barriers of TAM or CAF in solid tumor microenvironments, such as glioma, sarcomas, or Myeloid sarcomas.

    3.     The objective of the review is to discuss "B7-H3 in Pediatric Tumors: Far beyond Neuroblastoma" and emphasize the significance of utilizing B7-H3 as an immune-therapeutic target. However, the authors dedicate a significant amount of time to discussing each cancer type. For example, in section 2, paragraphs 1 to 3 focus on glioma; in section 3, paragraphs 1-3 discuss sarcoma; in section 4, paragraph 1 covers another cancer type. It would be more appropriate to focus on the hurdles or challenges related to the current therapy for these cancers.

    4.     Furthermore, these sections lack appropriate references, contain poorly written passages, and, in some cases, present incorrect statements. For example, the statement "Unlike OS and EwS, less is known about the genetic alterations associated with RMS" is incorrect, as there are several publications on the genomics and epigenetics of rhabdomyosarcoma. The authors should also discuss and present the expression of B7-H3 mRNA and protein across these cancers, as this data is publicly available in numerous databases.

    5.     In section 2, paragraph 4, it is necessary to clarify why the FDA halted the use of I131-omburtamab (8h9) in children with CNS or leptomeningeal metastases from neuroblastoma.

    6.     This manuscript should incorporate the advancements in current pre-clinical research on B7-H3 targeted therapy, as well as the advantages and disadvantages of different generations of B7-H3 CAR, bispecific CARs and armored CARs, to enhance the potency of B7-H3 CAR T therapy.

    7.     The authors should discuss the potential on target off tumor toxicities of B7-H3 targeted therapies, due to expression in normal tissues/organs

    8.     The detailed discussion of EGFRt does not align well with the scope of this review.

    9.     In the conclusion, paragraph 2, instead of introducing CD122-CD132 bi-specific mAbs or CD123 NKCE, the authors should summarize the research advances of B7-H3 directed immune therapies.

  1. There are numerous typos and errors in the English language. For instance, "life treating side effects" should be corrected to "life threatening," "T cel" should be "T cell," "On the other hands" should be "On the other hand," and "that of ERMS shows two picks" should be "that of ERMS shows two peaks."

Author Response

(The authors gave the same response as above.)

Reviewer 3 Report

The Review Article manuscript presented by Bottino et al. addresses the importance of B7-H3 immune checkpoint protein in paediatrics tumors. The review is timely and well written. Following are some suggestions and comments, mainly intended to incorporate small corrections and to improve the soundness of the manuscript:

1. The first paragraph in the introduction should include indication of other antibodies/studies that identified B7-H3 expressed on tumor cells, including antibody 376.96

2. Authors should include “paediatric” in the sentence “this review aims at discussing B7-H3 as a possible therapeutic target in tumors” in the last paragraph of the introduction (“…target in paediatric tumors”).

3. Authors must be consistent and choose uniform nomenclature throughout the manuscript (“B7-H3 CAR-T”, “B7-H3.CAR-T” or “B7-H3-CAR T”). It is suggested not to use the form “B7-H3.CAR-T” (used for most of CAR-T indications) because of the ambiguity with the dot which gets confused with the orthographic dot.

4. In the conclusion section, the authors start with mentioning Wilms tumor, which was not mentioned previous in the text. Could authors make a paragraph prior to conclusion with “other paediatric tumors”, which could include more paediatric cancers forms that as included in Table 1? In this novel section, authors could include relevant references, including the recent review PMID: 37110590

5. Authors should include a more informative figure legend to Figure 1, explaining the different forms of targeting B7-H3, as well as explaining the abbreviations used the Figure.

6. Title of Table 1 is “Currently active/recruiting”, but table also includes “completed” studies. This should be corrected.

7. To better put the readers in the appropriate context, authors could refer in the Introduction section to more comprehensive reviews on B7-H3 in cancer (not restricted to paediatric), such as PMID: 31099317 and 36859240

8. Other small corrections/typos:

-Line 18: the abbreviation “4Ig” stands for “4 immunoglobulin”. Please amend this.

- Line 28: the abbreviation mAb stands for “monoclonal antibodies”. Please amend this.

- Line 44: please write “4 immunoglobulin domains” instead of “4 Ig domains”.

Line 58: Please explain “B7-H3R/B7-H3 axis”

Line 72: Please amend abbreviation to “soluble B7-H3 (sB7-H3)”, as this is used later on.

- Line 117: the abbreviation “TAA” stands for “Tumor-associated antigens”. Please amend this.

- Line 120: The abbreviation “PD-1/PD-Ls”. “Ls” is ambiguous. Please amend this.

- Line 142: Please remove abbreviation “(ODAC)” as this is not further used.

- Line 148: delete “a” from “EGFRt is a non-immunogenic,…”

- Line 224: the expression “In the late 800’…” is ambiguous. Please amend this.

- Line 228: Please correct “TNF- , IFN- , and IL-1” to “TNF-α, IFN-γ, and IL1-β”.

- Line 229: Please correct “clear” to “partial”.

- Line 268: please remove “,” after “[48]”

- Line 289: Please correct “Y. Hu” to “Hu et al.”

- Line 281: please add punctuation after “[50]”

- Line 295: it is suggested to delete “only”

- Line 297: please amend the symbols. They also need amendment in lines 297, 331, 3336, 338

- Line 338: change “unwonted” by “unwanted”

- Line 370: it is suggested to change “realization” by “generation”

- Table includes nomenclature “CD276”. Please amend this.

Author Response

(The authors gave the same response as above.)

Round 2

Reviewer 1 Report

Very nice update after the edits.  Some minor comments.  A bit more English grammar and spelling updates (plural of "study" is "studies").  

Figure with cell therapies could stand a few more labels, about half of targets are labeled, but half are not making it a little confusing. 

Very last paragraph needs to be separated from acknowledgements.  

Very nice work overall and a nice paper. 

Minor grammar and spelling edits as above. 

Author Response

A bit more English grammar and spelling updates (plural of "study" is "studies").

Grammar and spelling have been updated

Figure with cell therapies could stand a few more labels, about half of targets are labeled, but half are not making it a little confusing. 

More labels have been added to both figures

Very last paragraph needs to be separated from acknowledgements.

The manuscript has been formatted by the Assistant Editor.